# Inconsistent Country-Wide Reporting of Adverse Drug Reactions to Antimicrobials in Sierra Leone (2017–2021): A Wake-Up Call to Improve Reporting

**DOI:** 10.3390/ijerph19063264

**Published:** 2022-03-10

**Authors:** Fawzi Thomas, Onome T. Abiri, James P. Komeh, Thomas A. Conteh, Abdulai Jawo Bah, Joseph Sam Kanu, Robert Terry, Arpine Abrahamyan, Pruthu Thekkur, Rony Zachariah

**Affiliations:** 1National Pharmacovigilance Center, Pharmacy Board of Sierra Leone, Freetown 02717, Sierra Leone; otabiri@pharmacyboard.gov.sl (O.T.A.); kompjames@yahoo.com (J.P.K.); contehansumusthomas@yahoo.com (T.A.C.); 2College of Medicine & Allied Health Sciences, University of Sierra Leone, Freetown 02717, Sierra Leone; abdulaijawobah@yahoo.com (A.J.B.); samjokanu@yahoo.com (J.S.K.); 3Institute of Global Health and Development, Queen Margaret University, Edinburgh EH21 6UU, UK; 4National Disease Surveillance Program, Directorate of Health Security and Emergencies, Ministry of Health and Sanitation, Freetown 02717, Sierra Leone; 5UNICEF, UNDP, World Bank, WHO Special Programme for Research and Training in Tropical Diseases (TDR), 1211 Geneva, Switzerland; terryr@who.int (R.T.); zachariahr@who.int (R.Z.); 6Tuberculosis Research and Prevention NGO (TB-RPC), Yerevan 0014, Armenia; arpine_abrahamyan@yahoo.com; 7Centre for Operational Research, International Union against Tuberculosis and Lung Disease, 75001 Paris, France; pruthu.tk@theunion.org

**Keywords:** SORT IT, operational research, VigiBase, health system strengthening, universal health coverage

## Abstract

Background: Monitoring of adverse drug reactions (ADRs) to antimicrobials is important, as they can cause life-threatening illness, permanent disabilities, and death. We assessed country-wide ADR reporting on antimicrobials and their outcomes. Methods: A cross-sectional study was conducted using individual case safety reports (ICSRs) entered into the national pharmacovigilance database (VigiFlow) during 2017–2021. Results: Of 566 ICSRs, inconsistent reporting was seen, with the highest reporting in 2017 and 2019 (mass drug campaigns for deworming), zero reporting in 2018 (reasons unknown), and only a handful in 2020 and 2021 (since COVID-19). Of 566 ICSRs, 90% were for antiparasitics (actively reported during mass campaigns), while the rest (passive reporting from health facilities) included 8% antibiotics, 7% antivirals, and 0.2% antifungals. In total, 90% of the reports took >30 days to be entered (median = 165; range 2–420 days), while 44% had <75% of all variables filled in (desired target = 100%). There were 10 serious ADRs, 18 drug withdrawals, and 60% of ADRs affected the gastrointestinal system. The patient outcomes (N-566) were: recovered (59.5%), recovering (35.5%), not recovered (1.4%), death (0.2%), and unknown (3.4%). There was no final ascertainment of ‘recovering’ outcomes. Conclusions: ADR reporting is inconsistent, with delays and incomplete data. This is a wake-up call for introducing active reporting and setting performance targets.

## 1. Introduction

An adverse drug reaction (ADR) is defined as “any response to a drug which is noxious and unintended that occurs at doses normally used in human beings for prophylaxis, diagnosis, therapy of disease, or for the modification of physiological functions [1]. Examples of ADRs include skin rashes, vomiting, diarrhea, and stomach ulcers. ADRs can result in life-threatening illness, permanent disabilities, and even death. The recognition of such reactions predicts a hazard for the future use of a specific drug(s) and may result in the institution of preventive measures, specific treatments, alteration of drug dosages, or even withdrawal of a given drug from the market [2].

Seminal research has demonstrated that ADRs are common in clinical practice. For example, in a study of 18,820 hospital admissions in the United Kingdom, 6.5% were due to ADRs, and 2.3% of those admitted died [3]. In the USA, a meta-analysis of 39 prospective studies of hospitalized patients showed an incidence of serious ADRs of 6.7% and of fatal ADRs of 0.32%. This makes ADRs among the top ten causes of death in the USA [4]. On a worldwide level, a meta-analysis of observational studies showed that 4.9%, to as high as 41.3% of hospital admissions, are due to ADRs, with considerable economic implications [5,6].

Considering the importance of ADRs, the World Health Organization (WHO) set up an international program for drug monitoring [7]. To date, there are 136 countries that share their data with VigiBase, the WHO global database of individual case safety reports (ICSRs). An ICSR includes information on adverse events, problems related to drugs, and complaints filed by consumers with respect to any given drug [8]. This database is maintained by the Uppsala Monitoring Center, which is a WHO Collaborating Center for International Drug Monitoring based in Sweden [9]. VigiBase is the single largest drug safety repository in the world. At the country level, a pharmacovigilance database called VigiFlow facilitates the standardized collection, processing, and sharing of ICSR data to facilitate data analysis. VigiFlow feeds the VigiBase (Figure 1).

Sierra Leone has VigiFlow, which was set up in 2007 and is managed by the National Pharmacovigilance Centre housed at The Pharmacy Board of Sierra Leone; this also serves as the National Medicines Regulatory Authority. There are ongoing efforts to improve pharmacovigilance in Sierra Leone, including making resources available for the establishment of a well-organized system for monitoring and evaluating ADRs [10]. As we are in an era of growing antimicrobial resistance (AMR), pharmacovigilance of second and third-line antimicrobials has become particularly important. In addition, newer generation antimicrobials are also coming on the market and need to be monitored carefully. This is particularly relevant to Africa, where the majority of ADRs are related to antibiotics, and bacterial infections are still a leading cause of death [11,12].

Globally, 3,446,696 reports for antimicrobials were submitted to VigiBase, of which 130,000 were from Africa for 2022 [13].

Sierra Leone has a high burden of infectious diseases where local patterns of disease, prescription practices, and the use of over-the-counter and counterfeit (or adulterated) antimicrobials might influence the incidence of ADRs [11,14].

The availability of the VigiFlow database in Sierra Leone provides an excellent opportunity to assess ADRs from a country-wide perspective. Such information could guide drug safety, drug regulation systems, clinical practice, and public health programming. Further, such an analysis would provide valuable insight into ‘what works and what needs to be improved in relation to reporting to VigiFlow. A PubMed search found a few studies from Nigeria [15], Ghana [16], and Togo [17] that assessed ADRs at a country-wide level. Only one study from Zimbabwe assessed the performance of the pharmacovigilance reporting [18]. There have been no studies from Sierra Leone assessing the quality and performance of pharmacovigilance reporting. The only study conducted in the country focused on resources and programmatic aspects for implementing pharmacovigilance [10].

We, thus, aimed to assess the reporting of ADRs to antimicrobials, their management, and outcomes from public health facilities in Sierra Leone. At a country-wide level, between 2017–2021, our specific objectives were to assess in the VigiFlow database: (a) trends in the total numbers of individual case safety reports and ADRs related to antimicrobials, (b) timeliness and completeness of reporting, (c) seriousness and actions taken for ADRs, (d) types of ADRs using system organ classification, and (e) patient outcomes stratified by antimicrobial drug classes.

## 2. Materials and Methods

### 2.1. Study Design

A cross-sectional study of individual case safety reports routinely entered into the national pharmacovigilance database (VigiFlow).

### 2.2. Setting

#### 2.2.1. General Setting

Sierra Leone is situated on the coast of West Africa with a population of about 8 million, of which at least 1 million live in the capital city of Freetown [19]. The country has an area of 71,740 km^2^ and is divided into five administrative regions: the Northern Region, North-Western Region, Eastern Region, Southern Region, and the Western Area. The five areas are subdivided into 16 districts with 29 public hospitals.

#### 2.2.2. Specific Setting

The National Pharmacovigilance Center and ADR reporting.

The Pharmacy Board of Sierra Leone hosts the National Pharmacovigilance Centre, which is mandated to monitor drug safety by the Pharmacy and Drug Act of 2001 [20]. The center was established in 2007 and received country-wide reports of suspected ADRs. In addition, all 29 public hospitals in Sierra Leone are pharmacovigilance reporting sites and have a pharmacovigilance focal point. All these focal points are pharmacists who have been trained on pharmacovigilance and drug safety issues.

There are passive and active approaches to ADR reporting in the country. Reporting from health facilities is passive (voluntary), while during mass drug campaigns there is active reporting integrated as part of the campaign. The information flow starts when a patient or drug consumer experiences a suspected ADR (Figure 1). Filling out the information in the standardized ADR reporting form (Figure A1) can happen via three channels: (a) the attending healthcare provider may spontaneously fill out the ADR report, (b) it may be delegated to one of the designated pharmacovigilance focal persons, and (c) any individual inside or outside the health system can access a web link for reporting using an electronic ADR form by using their phones or computers. The transmission of ADR reporting forms to the pharmacovigilance center where VigiFlow is located can also happen via three channels: (a) ADR reporting forms can be directly transmitted by health workers, (b) they can be collected by field supervisory teams who visit on an unscheduled and random basis, and (c) electronic ADR forms can feed directly into VigiFlow. This is the most rapid and efficient reporting route.

Processing of Individual’ Case Safety Reports (ICSR) in VigiFlow.

All ADR forms (paper-based and electronic) are cross-checked by central pharmacovigilance staff. A completeness check is conducted on whether (or not) a set of 12 required variables have been filled in. These are patient identification, sex, age, name of suspected drug, strength, dose, the start date of administration, description of the reaction, therapeutic indication, onset time of the reaction, outcome, and reporters’ details.

If any of these variables are missing, a follow-up inquiry is made but at the discretion of the pharmacovigilance staff. When the form is deemed complete, an acknowledgement note is sent back to the reporter, indicating that the ADR report was received. All suspected ADRs forms are entered into VigiFlow and are thereafter considered ICSRs.

All ICSRs in VigiFlow are routinely reviewed for signal detection purposes when the drug reaction is serious and/or is not listed in the patient information leaflet or the summary of product characteristics.

A serious ADR is defined as one that results in death, is life-threatening, requires hospitalization or prolongation of existing hospitalization, results in persistent or significant disability/incapacity, or is a congenital anomaly/birth defect [21]. A ‘signal’ refers to reported information on a possible causal relationship between a suspected ADR and a drug, the relationship being unknown or incompletely documented previously. This causal relationship assessment is conducted according to the WHO, UMC methodology, or the Naranjo algorithm [22]. Feedback of any causality assessment is sent to the reporter.

Drugs and therapeutic committees are being set up at pharmacovigilance sites to help coordinate pharmacovigilance activities.

### 2.3. Study Population and Period

This study included all individual case safety reports available in the pharmacovigilance database (VigiFlow) from January 2017 to September 2021.

### 2.4. Data Source, Collection, and Validation

All data variables were extracted from the VigiFlow database and were received country-wide. The variables included identifiers, gender, age, antimicrobial class and drug names, types of reported ADRs, seriousness, actions taken, and patient outcomes. All ICSR data were cross-validated as part of the routine pharmacovigilance procedures prior to entry into VigiFlow.

### 2.5. Statistical Analysis

The ICSR data were exported to Microsoft Excel and analyzed using SPSS version 21 (International Business Machines Corporation, SPSS Statistics GradPack and Faculty Packs, New York, NY, USA). ADRs were categorized using the medical dictionary for regulatory activities (MedDRA) classification and presented in the system organ classification (SOC) [23]. Patient outcomes were stratified by antimicrobial class and AWaRe (access, watch, and reserve) categories for antibiotics. The “access” category includes first- and second-line antibiotics for common infections. The “watch” category includes antibiotics with a higher potential to develop resistance, while the “reserve” category is restricted to antibiotics for multidrug-resistant bacterial infections. For the purposes of this study, all suspected ADRs were considered as ADRs. The time taken to report an ADR (ADR reporting time) was calculated as the time from the date of onset of symptoms to the date of entry into the VigiFlow database. The completeness score was auto-calculated by the VigiFlow for each ICSR based on the completeness in reporting the required variables (the calculated score lies between 0 to 100%).

We used a descriptive analysis. Frequencies and proportions were used to report the distribution of ADR types and patient outcomes. In addition, median and interquartile ranges (IQR) were used to summarize ages and ADR reporting times.

## 3. Results

Of 568 ICSRs reported, 2 were for drugs that were not antimicrobials and were excluded from the analysis. Of the remaining 566 reports involving antimicrobial drugs, 318 (56%) were for females, and the median overall age of the patients was 29 years (IQR 18, range 2–82).

### 3.1. Trends in Reporting of ICSRs and ADRs

Figure 2 shows the trends in total numbers of ICSRs and ADRs related to antimicrobial drugs. There was an inconsistent reporting trend, with the highest number of reports in 2017 and 2019, which coincided with active reporting during mass drug campaigns for deworming using Ivermectin and Albendazole (2017 being country-wide and 2019 involving the western area). For voluntary reports from public health facilities, there was zero reporting in 2018 (reasons unknown) and only a handful of reports in 2020 and 2021 (COVID-19 pandemic).

### 3.2. Timeliness and Completeness of Reporting

Table 1 shows the timeliness and completeness of reporting to the VigiFlow. There were major delays in ADR reporting, with 90% of reports taking more than 30 days and 39% taking over 180 days to be entered into VigiFlow (median = 165 days; range 2–420 days). In total, 240 (43%) of 566 ICSRs were 100% complete and 44% of the ICSRs had less than 75% of the required variables filled in (desired target = 100%). The majority (64%) of reports were made by pharmacists, followed by nurses and allied health workers (23%).

### 3.3. Seriousness and Actions Taken for Antimicrobial-Related ADRs

There were 10 serious ICSRs that reported ADRs related to kanamycin (3), albendazole (2), ivermectin (1), trimethoprim-sulfamethoxazole (1), cycloserine (1), streptomycin (1), and antiretrovirals (1). The serious ADRs experienced in these 10 cases include deafness, jaundice, balance disorder, vomiting, erectile dysfunction, oropharyngeal pain, loss of consciousness, tinnitus, dyspnea, and body rash. Data on the action taken was not recorded in 37% of the reports. In 49%, the action was deemed ‘not applicable’, as they involved single-dose drugs (such as ivermectin) as part of mass campaigns.

In 18 individuals, the suspected antimicrobial (one or more) was withdrawn as a result of the adverse drug reaction experienced. (Table 2). These included: artesunate/amodiaquine, isoniazid, trimethoprim-sulfamethoxazole, ampicillin, chloramphenicol, quinine, artemether/lumefantrine, gentamicin, lamivudine, nevirapine, zidovudine, benzathine penicillin, amoxicillin/clavulanic acid, doxycycline, erythromycin, ivermectin, ethambutol, pyrazinamide, rifampicin, and albendazole.

### 3.4. Types of Antimicrobial-Related ADRs

Among individuals (*n* = 566), the most commonly reported ADRs included the gastrointestinal system (60%), nervous system (30%), general disorders and administration site conditions (26%), and skin and subcutaneous tissues (19%) (Table 3). ADRs with prevalence >10% included: diarrhea (25%), abdominal pain (16%), headache (18%), pruritus (13%), vomiting (11%), and dizziness (11%).

### 3.5. Patient Outcomes Stratified by Antimicrobial Drug Classes

Of 566 ICSR reports, 90% were for antiparasitics (actively reported during mass campaigns), while the rest (voluntarily reported from health facilities) included 8% antibiotics, 7% antivirals, and 0.2% antifungals. There was one death, and this was related to the anti-parasitic drugs used during a mass drug distribution campaign (Ivermectin and Albendazole) (Table 4). Overall, 36% of patient outcomes were classified as recovering, with no final ascertained patient outcome available. The picture was similar for antiparasitics (35%), where despite the last drug administered dated during a mass campaign of 2019 (2 years prior to the census date for the data analysis in this study). The patient outcome remained stagnant at ‘recovering’, with no further ascertainment of patient outcomes.

## 4. Discussion

This first country-wide study from Sierra Leone shows inconsistent reporting trends, with the highest reporting in 2017 and 2019 (during mass drug campaigns for deworming), zero reporting in 2018 (reasons unknown), and only a handful of reports in 2020 and 2021 (since COVID-19). The majority (90%) of reports were for antiparasitics (actively reported in mass campaigns), while a small number of reports were a result of passive (voluntary) reporting from health facilities. In addition, shortcomings were seen in the timeliness and completeness of ADR reports and the ascertainment of patient outcomes.

The findings of this study are important, as they serve as a wake-up call (an opportunity) to improve antimicrobial pharmacovigilance in the country. The data also highlight the potential impact that an ‘active approach’ to ADR reporting can have on enhancing reporting from health facilities, as was evident during the mass drug campaigns. Furthermore, the recent revelation of five million annual global AMR deaths, with a higher risk in sub-Saharan African countries such as Sierra Leone [12], provides an imperative to improve access to antibiotics at lower levels of the health pyramid. Such efforts would need to be accompanied by high standards of monitoring to ensure the safe administration of antimicrobials to communities.

The study strengths are that: (a) we included country-wide data, and the findings are likely to reflect the operational reality on the ground, (b) we used data from the standardized VigiFlow database, which can serve as a baseline for future assessments and for building synergies between routine monitoring and operational research, and (c) we responded to an identified national operational research priority, increasing the likelihood for research uptake. We also adhered to the STROBE guidelines statement for the reporting of observational studies in epidemiology [24]. In line with STROBE, there is a summary of what was conducted and what was found, in terms of a background and rationale, methods, results, and discussion.

A study limitation is that we used ‘time from ADR symptoms’ to calculate the reporting time, unlike the usual practice of ‘time from notification in the health system’. Although this may exaggerate the reporting time, in light of the magnitude of delays in our study, it is unlikely to have any major effect on the delays. Furthermore, we classified all suspected ADRs as ADRs. In countries such as Sierra Leone, which are understandably facing teething problems in establishing robust ADR reporting systems, this is unavoidable. It has also been the case in other settings in Africa [14,18,25]. However, as we move forwards to improve the reporting and follow up systems, we expect to be better able to confirm the implicated drugs(s) for suspected ADRs (establish causality). Finally, we had missing records for the seriousness of ADRs, which may impact the overall picture of serious ADRs.

The study findings have a number of policy and practice implications. First, the current system of ADR reporting is designated as a purely voluntary and passive exercise that is perceived as lying largely within the domain of pharmacists. There are also multiple channels for filling out and transmitting pharmacovigilance data, which tends to complicate and weaken the reporting system. In Sierra Leone, where health workers are often overloaded, there are downsides to such a passive approach. For example, while health workers might be recognizing and managing ADRs in their daily clinical work, they may not be motivated to report due to overriding priorities. Despite being long, the median delay in our study (165 days) was still better than that seen in Uganda (341 days, [25]) and Zimbabwe (548 days, [18]).

Health workers may also entertain the perception that pharmacovigilance is a pharmacist driven activity, resulting in the delegation of reporting responsibility, as was evident in our study. In total, 64% of reports were made by pharmacists [14,25]. The resulting low reporting from health facilities is, in essence, the ‘tip of the iceberg’. There were only two ADRs reported for non-antimicrobial drugs during five years. In addition, the fact that we had only 566 ICSRs (for antimicrobials) over five years, implying that only about 110 cases were reported per year for a population of eight million, provides some evidence to support that thinking. The encouraging scenario seen during mass drug campaigns where ‘active ADR reporting’ was practiced provides an eye-opener to what could change if compulsory reporting was introduced. Comparative studies in ADR reporting in tuberculosis in a tertiary hospital in India [26] and from Burkina Faso (malaria) [27] showed up to a five-fold increase in reporting when an active approach was implemented.

Possible ways forwards for changing this paradigm include: (a) reinforcing training on pharmacovigilance among health workers, (b) introducing compulsory ADR reporting at all health facilities, (c) enforcing performance targets stipulated in the national guidelines [21], such as the seven day limit for reporting serious ADRs and a 28 day limit for other ADRs, (d) ensuring a completeness score of ≥95% for filling-in all variables in ADR forms, and (e) shifting to the already existing free online electronic ADR reporting system using mobile phones and/or computers. “Before and after” research to measure the impact of these changes would benefit evidence-informed decision making.

Second, a considerable proportion (36%) of individuals with ADRs were classified as ‘recovering’ even when the last reported ADR dated back to 2019 (two years prior to this analysis). This indicates that ascertainment of the final patient outcomes is not happening. This shortcoming is attributed to the fact that VigiFlow was primarily designed to provide ‘signals’ of new ADRs upwards to the central level. Similar findings and attributions were also reported from Zimbabwe [18]. It is important that this improves through a system of follow up for all recovering patients, as ascertaining the final patient outcomes is vital for decision making from a programmatic perspective. For example, the ascertained number of ADR-related deaths at health facilities is a negative indicator of patient safety.

A way forward is to maintain a copy of the preliminary ADR report as part of the electronic or paper-based records in each health facility and update the record after ascertainment of patient outcomes. This information can then be transmitted to the central level electronically every 28 days.

Finally, despite the low reporting, we identified 18 drugs that had to be withdrawn due to ADRs, and there were 10 serious drug reactions and 1 associated death. These numbers are probably an underestimate, and they add to the ample justification for a wake-up call to improve the reporting system. In addition, several ADRs (e.g., diarrhea, itching, etc.) had a prevalence of >10% and reached as high as 25% in some instances. Some drugs, including the quinolones, aminoglycosides, and antiretroviral drugs, were shown to cause serious adverse drug reactions. Such data would be useful to inform the training needs on managing ADRs at all levels of the health pyramid.

## 5. Conclusions

This first county-wide assessment of the ADR reporting system in Sierra Leone shows inconsistent reporting, with delays and incomplete data. This is a wake-up call for introducing compulsory (active) reporting and setting performance targets. As Sierra Leone is currently updating its guidelines and standard operating procedures (SOPs) for pharmacovigilance, and also the imminent deployment of the Med Safety app in collaboration with the WHO, UMC, and Medicines and Healthcare Products Regulatory Agency UK (MHRA), there is an opportunity to change the paradigm and take on some of these recommendations.

## Figures and Tables

**Figure 1 ijerph-19-03264-f001:**
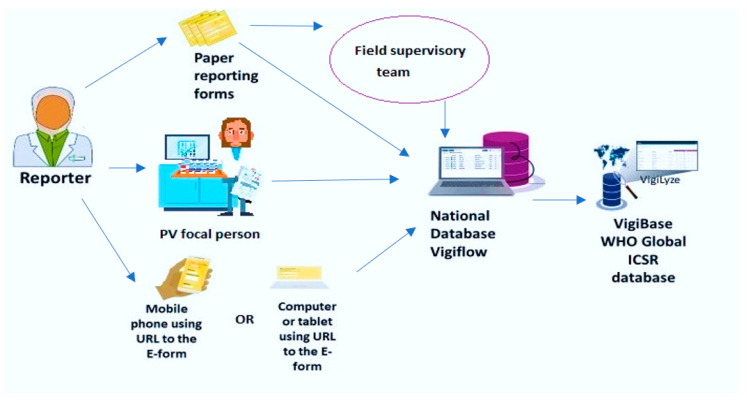
Information flow for the reporting of adverse drug reactions to VigiFlow in Sierra Leone and the VigiBase (World Health Organization).

**Figure 2 ijerph-19-03264-f002:**
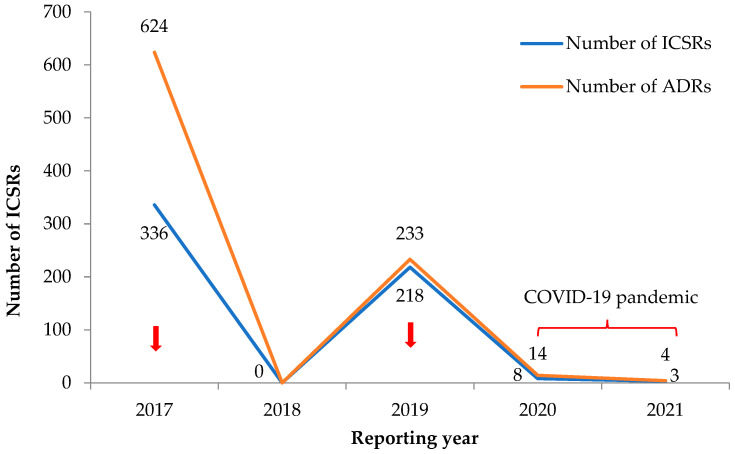
Yearly trend in the number of reported ICSRs with ADRs related to antimicrobials in the VigiFlow between 2017 and 2021 in Sierra Leone. Mass drug campaigns: country-wide in 2017 and in the western regions in 2019. Abbreviations: ADRs, adverse drug reactions; ICSR, individual case safety report. Note: An ICSR is reported per individual and could have multiple ADRs. Thus, the number of ADRs may be higher than the number of reported ICSRs.

**Table 1 ijerph-19-03264-t001:** Timeliness and completeness of ICSR reporting (*n* = 566) in the VigiFlow in Sierra Leone during 2017–2021.

Characteristics	*n*	(%)
Total	566	
Time to reporting (in days) ^1^		
<30	56	(9.9)
30–180	289	(51.1)
≥180	221	(39.0)
Completeness score ^2^		
0–25%	2	(0.4)
26–50%	28	(4.9)
51–75%	221	(39.0)
76–99%	21	(3.7)
100%	240	(42.4)
Not recorded	54	(9.5)
Reported by		
Pharmacists	364	(64.3)
Nurses and allied health workers ^3^	129	(22.8)
Physicians	3	(0.5)
Consumers or non-health professionals	17	(3.1)
Not recorded	53	(9.4)

^1^ Duration between the start of ADR (symptom onset) and entry into VigiFlow. ^2^ Completeness score is auto-calculated by the VigiFlow for each ICSR based on the proportion of variables in the ADR form correctly filled in (score lies between 0 to 100%). ^3^ Includes community health officers and nurse assistants. Abbreviation: ICSR, individual case safety report.

**Table 2 ijerph-19-03264-t002:** Seriousness and actions taken for antimicrobial-related ADRs in ICSRs (*n* = 566) reported to the VigiFlow between 2017 and 2021 Sierra Leone.

Characteristics	*n*	(%)
Total	566	
Serious reaction ^1^		
Yes	10	(1.8)
No	501	(88.5)
Not recorded	55	(9.7)
Action taken ^2^		
Drug withdrawn	18	(3.2)
Dose increased	1	(0.2)
Dose reduced	2	(0.4)
Unknown	56	(9.9)
Dose not changed	4	(0.7)
Not applicable	277	(48.9)
Not recorded	208	(36.7)

Abbreviation: ADR, adverse drug reaction. ^1^ A serious reaction is one that is life-threatening, requires hospitalization or prolongation of existing hospitalization, results in persistent or significant disability/incapacity, or is a congenital anomaly/birth defect, or results in death. ^2^ Unknown implies no knowledge of whether any action taken, dose not changed implies the initial dose was maintained, not applicable implies the drug was single-dose and no further action was to be taken (involved drugs dispensed through mass campaigns), and not recorded implies unfilled data.

**Table 3 ijerph-19-03264-t003:** Types of antimicrobial-related ADRs in individuals (*n* = 566) according to the MedDRA system organ classification reported to the VigiFlow in Sierra Leone during 2017–2021.

Adverse Drug Reactions	*n*	(%) ^1^
Gastrointestinal disorders	337	(59.5)
Diarrhea	143	(25.3)
Abdominal pain	93	(16.4)
Vomiting	61	(10.8)
Nausea	30	(5.3)
Others (loss of appetite, dysphagia, loss of taste, increased appetite, and dyspepsia)	10	(1.8)
Nervous system disorders	167	(29.5)
Headache	101	(17.8)
Dizziness	62	(11.0)
Seizure	3	(0.5)
Peripheral neuropathy	1	(0.2)
General disorders and administration site conditions	130	(23)
Asthenia/Malaise/Fatigue	62	(11)
Generalized, pelvic, and musculoskeletal pain	30	(5.3)
Fever	26	(4.6)
Pedal edema/Facial edema/Periorbital edema	12	(2.1)
Skin and subcutaneous tissue disorders	98	(17.3)
Pruritus	72	(12.7)
Skin rashes	21	(3.7)
Others (Increased sweating, erythema, and urticaria)	5	(0.9)
Psychiatric disorders (insomnia, confusion, somnolence, anorexia, abnormal gait, hallucination, and nightmares)	29	(5.1)
Musculoskeletal and connective tissue disorders (arthralgia, muscle pain, and muscle weakness)	16	(2.8)
Ear and labyrinth disorders (tinnitus, hearing impairment, and others)	16	(2.8)
Eye disorders (blurred vision, conjunctivitis, and others)	11	(1.9)
Renal and urinary disorders (polyuria, urinary urgency, and acute renal injury)	8	(1.4)
Cardiac disorders (palpitations and orthostatic hypotension)	7	(1.2)
Respiratory, thoracic, and mediastinal disorders (coughing, chest discomfort, and dyspnea)	7	(1.2)
Immune system disorders (allergy and hypersensitivity)	6	(1.1)
Reproductive system and breast disorders (menorrhagia, erectile dysfunction, and others)	6	(1.1)
Blood and lymphatic system disorders (bleeding and anemia)	4	(0.7)

^1^ Individuals can have multiple ADRs, and, thus, the percentages do not add up to 100%.

**Table 4 ijerph-19-03264-t004:** Patient outcomes stratified by antimicrobial drug classes in ICSRs in the VigiFlow between 2017–2021 in Sierra Leone.

Antimicrobial Class	Recovered		Recovering		Not Recovered		Death		Unknown		Total
	*n*	(%)	*n*	(%)	*n*	(%)	*n*	(%)	*n*	(%)	N
Antiparasitics	313	(62.5)	180	(35.4)	4	(0.8)	1	(0.2)	16	(2.2)	514
Antibiotics	21	(47.8)	17	(38.6)	3	(6.8)	0	(0)	3	(6.8)	44
Access	8		10		0		0		2		20
Watch	12		6		2		0		1		21
Reserve	1		1		1		0		0		3
Antiviral	3	(42.9)	4	(57.1)	0	(0)	0	(0)	0	(0)	7
Antifungals	0	(0)	0	(0)	1	(10.0)	0	(0)	0	(0)	1
Total	337	(59.5)	201	(35.5)	8	(1.4)	1	(0.2)	19	(3.4)	566

## Data Availability

Requests to access these data should be sent to the corresponding author.

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
