# Peer review of "Inconsistent Country-Wide Reporting of Adverse Drug Reactions to Antimicrobials in Sierra Leone (2017–2021): A Wake-Up Call to Improve Reporting"

_ijerph, 2022, doi:10.3390/ijerph19063264_

Round 1

Reviewer 1 Report

The manuscript reports inconsistent country-wide reporting of adverse drug reactions to antimicrobials in Sierra Leone (2017-2021). Monitoring of adverse drug reactions (ADRs) to antimicrobials is important  topic. The countrywide ADR reporting on antimicrobials, and their outcomes were presented. A cross-sectional study of individual case safety reports (ICSRs) entered into the national pharmacovigilance database (Vigi-Flow) during 2017-2021. Was performed.   Of 566 ICSRs, inconsistent reporting was seen, with highest reporting in 2017 and 2019, zero reporting in 2018, and only a handful in 2020 and 2021. There were 10 serious ADRs, 18 drug withdrawals and 60% ADRs affected the gastrointestinal system. ADR reporting is inconsistent with delays and incomplete data that could promote introducing active reporting and setting performance targets. The manuscript is interesting, reporting relevant issues and I have few minor remarks.

At the endo of introduction part there is need to clearly state the novelty of performed research. There is need to include the research hypothesis and how is it to be tested in the research.

Lines 214-216: “There were 10 serious ICSRs that reported ADRs related to Kanamycin (3), Albendazole (2), Ivermectin (1), Trimethoprim-sulfamethoxazole (1), Cycloserine (1), Streptomycin (1), and antiretrovirals (1)”. There is need to report what were those serious ICSRs.

Lines 219-224: “In 18 individuals the suspected antimicrobial (one or more) was withdrawn”. Can the authors in short report why they were withdrawn?

Author Response

REVIEWER 1

  1. At the end of the introduction part, there is a need to clearly state the novelty of performed research. There is need to include the research hypothesis and how is it to be tested in the study.

Response

This is a cross-sectional study that did not utilize any inferential statistics and p-value, and also, there was no analysis of  cause and effect.

  1. Lines 214-216: "There were 10 serious ICSRs that reported ADRs related to Kanamycin (3), Albendazole (2), Ivermectin (1), Trimethoprim-sulfamethoxazole (1), Cycloserine (1), Streptomycin (1), and antiretrovirals (1)". There is a need to report what were those serious ICSRs.

Response

The serious ADRs experienced from the 10 cases include deafness, jaundice, balance disorder, vomiting, erectile dysfunction, oropharyngeal pain, loss of consciousness, tinnitus, dyspnea and body rash. This has been added to the manuscript

  1. Lines 219-224: "In 18 individuals, the suspected antimicrobial (one or more) was withdrawn". Can the authors, in short report why they were withdrawn?

Response

They were withdrawn because they reported adverse drug reactions, including some serious ADR. This had been added to the manuscript and was also mentioned in the discussions.

Reviewer 2 Report

Dear Author:

1. The sample size appears to be small, despite a large number of patients experiencing adverse antimicrobial agent reactions between 2017 and 2021.

2. Why do you believe the number was zero in 2018?

3. Why was the ADR of antiviral agent not listed during the COVID-19 pandemic?

4. I propose adding a paragraph about the most recent WHO ADR statistics.

5. I propose categorizing Table 3 as antiviral, antibiotic, antiparasitic, or antifungal.

6. At the end of the discussion section, I propose including a list of antimicrobial agents that have lethal adverse reactions.

7. Minor editing changes are required for the manuscript.

Just a quick note: leave space between the text and the reference. Lines such as 53 USA[4] and 57 monitoring[7] are examples. Monitoring [7] and the United States [4].

Author Response

REVIEWER 2

  1. The sample size appears to be small, despite a large number of patients experiencing adverse antimicrobial agent reactions between 2017 and 2021.

Response

In this regard, the sample that was utilized was based on the number of ICSRs received and used in this study for the stipulated period of 2017 to 2021.

  1. Why do you believe the number was zero in 2018?

Response

In 2018 there were no reports for antimicrobials, although there were reports for other drug classes, which this paper did not cover. Reporting ADRs is voluntary or spontaneous, and the high number of reports for 2017 and 2019 was campaign driven and has been mentioned.

  1. Why was the ADR of antiviral agents not listed during the COVID-19 pandemic?

Response

The national pharmacovigilance centre did not receive ADRs from any antiviral therapeutic agents. The National Medicine Regulatory agency in Sierra Leone did not receive any application or authorize any specific Covid-19 antiviral drug for use in Sierra Leone. However, adverse events following immunization for Covid-19 vaccines were received and evaluated but are outside the scope of this study. For this study, we concentrated on therapeutics and not vaccines that are used for the Covid -19 Pandemic

  1. I propose adding a paragraph about the most recent WHO ADR statistics.

Response

So I went to Vigibase and extracted the number of reports of adverse drug reactions due to antimicrobials globally and in Africa. For Africa 130, 000 reports for antimicrobials, and globally 3, 446,696 – reference Uppsala monitoring centre (UMC) adverse drug reaction reporting statistics 2022- This has been added to the manuscript

  1. I propose categorizing Table 3 as antiviral, antibiotic, antiparasitic, or antifungal.

Response

So we decided to be consistent in presenting this information based on MedDRA, System organ classification and preferred terms. This is also in line with other studies in Africa and globally that have been referenced in this paper

  1. At the end of the discussion section, I propose including a list of antimicrobial agents that have lethal adverse reactions.

Response

Some antimicrobial that can result in serious adverse drug reactions include aminoglycoside (kanamycin, streptomycin), antivirals (tenofovir, efavirenz, lamivudine), quinolones (ciprofloxacin)

This has been added to the manuscript

  1. Minor editing changes are required for the manuscript.

Just a quick note: leave space between the text and the reference. Lines such as 53 USA[4] and 57 monitoring[7] are examples. Monitoring [7] and the United States [4].

Response

This has been done in the manuscript

Reviewer 3 Report

Attached below.

Author Response

REVIEWER 3

  1. Title should be edited. The title has an exclamation mark, making it less scientific.

Response

               The exclamation mark has been removed from the manuscript

  1. The experiments designed and performed by the authors to confirm their hypothesis are simple and fairly conclusive. However, the statistical analysis is descriptive (Line 178). With no explanation on why this is the case. Statistical significance tests performed will help in strengthening the validity of data obtained.

Response

Thank you for your suggestion. However, the statistical analyses were informed by this study's aim and objectives and based on an extensive literature review. As this is the first such study done in Sierra Leone, we believe this will provide baseline data for future studies, including hypotheses testing and inferential statistics. We will add this to the study limitation.

  1. Lines 172-173 unclear – For the purpose of this study, all suspected ADRs were considered as ADRs. What are the authors implying here?

Response

So, all adverse drug reactions are usually suspected until a causal relationship is established. Not all of the suspected ADRs have an established causal link and therefore

  1. The paper does not explain the reason for delays in ADR reporting (Line 202) and implies overburdening of the staff (Line 293, 294) as a probable reason. More insight should be given here with information from actual health offices to improve the methodologies for the future.

Response

One reason for low reporting is that it is voluntary and not mandatory, as with industries. A feasibility study was done during one of our training, and health professionals said they don't report because of fear of litigation. In some countries, Marketing Authorization Holders are mandated to report, which contributes significantly to amount of reports received. We have updated our pharmacovigilance regulation and are in the process of ensuring that marketing authorization holders take liability for their products on our market by establishing pharmacovigilance systems to monitor and reports ADRs.

  1. Line 276- STROBE guidelines statement should be explained here, at least a summary of the same.

Response

The study has a summary of what was done and what was found in terms of having an introduction (scientific background and rationale), methods (key elements of the study design and statistical analysis), results and discussion and other relevant information

This has been added to the manuscript

  1. Line 214 – talks about serious ICSR. What qualifies as a serious ICSR is not explained in depth in the paper.

Response

Serious ICSRs are reported ADR cases that result in death, are life-threatening, require hospitalization or prolong existing hospitalization, result in persistent or significant disability/incapacity, or a congenital anomaly/birth defect. Explanation of an ICSR was done in the introduction (reference 8). The results section also explained a serious ADR, table 2 as a footnote.

  1. What measures should be taken to keep a check on "recovering" patients was not explained in the paper- Line 34, as that was one of the flaws the authors determined in the reporting system.

Response

This has been addressed in the manuscript by proposing a follow up system for all recovering patients

This has been addressed in the manuscript